# Anti-SARS-CoV-2 IgG against the S Protein: A Comparison of BNT162b2, mRNA-1273, ChAdOx1 nCoV-2019 and Ad26.COV2.S Vaccines

**DOI:** 10.3390/vaccines10010099

**Published:** 2022-01-10

**Authors:** Joanna Szczepanek, Monika Skorupa, Agnieszka Goroncy, Joanna Jarkiewicz-Tretyn, Aleksandra Wypych, Dorota Sandomierz, Aleksander Jarkiewicz-Tretyn, Joanna Dejewska, Karolina Ciechanowska, Krzysztof Pałgan, Paweł Rajewski, Andrzej Tretyn

**Affiliations:** 1Centre for Modern Interdisciplinary Technologies, Nicolaus Copernicus University, 87-100 Torun, Poland; monika_skorupa@umk.pl (M.S.); wypych@umk.pl (A.W.); 2Faculty of Biological and Veterinary Sciences, Nicolaus Copernicus University, 87-100 Torun, Poland; prat@umk.pl; 3Faculty of Mathematics and Computer Science, Nicolaus Copernicus University, 87-100 Torun, Poland; gemini@mat.umk.pl; 4Cancer Genetics Laboratory Ltd., 87-100 Torun, Poland; prezes@genetykatorun.pl (J.J.-T.); dorota.sandomierz@gmail.com (D.S.); aleksanderjt@gmail.com (A.J.-T.); jdejewska@genetykatorun.pl (J.D.); karolinaciech@wp.pl (K.C.); 5Academic Research Center AKAMED Ltd., 87-100 Torun, Poland; 6Polish-Japanese Academy of Information Technology, 02-008 Warszawa, Poland; 7Department of Allergology, Clinical Immunology and Internal Diseases, Collegium Medicum, Nicolaus Copernicus University, 85-067 Bydgoszcz, Poland; palgank@cm.umk.pl; 8Department of Internal and Infectious Diseases, Provincial Infectious Disease Hospital, 85-067 Bydgoszcz, Poland; rajson@wp.pl

**Keywords:** SARS-CoV-2, COVID-19, Pfizer/BioNTech, Oxford/AstraZeneca, Moderna, Janssen/Johnson & Johnson, vaccines, antibodies

## Abstract

Background: COVID-19 vaccines induce a differentiated humoral and cellular response, and one of the comparable parameters of the vaccine response is the determination of IgG antibodies. Materials and Methods: Concentrations of IgG anti-SARS-CoV-2 antibodies were analyzed at three time points (at the beginning of May, at the end of June and at the end of September). Serum samples were obtained from 954 employees of the Nicolaus Copernicus University in Toruń (a total of three samples each were obtained from 511 vaccinated participants). IgG antibody concentrations were determined by enzyme immunoassay. The statistical analysis included comparisons between vaccines, between convalescents and COVID-19 non-patients, between individual measurements and included the gender, age and blood groups of participants. Results: There were significant differences in antibody levels between mRNA and vector vaccines. People vaccinated with mRNA-1273 achieved the highest levels of antibodies, regardless of the time since full vaccination. People vaccinated with ChAdOx1 nCoV-2019 produced several times lower antibody levels compared to the mRNA vaccines, while the antibody levels were more stable. In the case of each of the vaccines, the factor having the strongest impact on the level and stability of the IgG antibody titers was previous SARS-CoV-2 infection. There were no significant correlations with age, gender and blood type. Summary: mRNA vaccines induce a stronger humoral response of the immune system with the fastest loss of antibodies over time.

## 1. Introduction

Coronavirus disease 2019 (COVID-19), which is caused by β-coronavirus SARS-CoV-2, surfaced in December of 2019 and has since become a global pandemic. The disease rapidly progressed and caused many deaths, forcing scientists to create a safe and effective tool to prevent it. Vaccines are the most promising tool for solving this global problem [1]. In Poland, vaccination against the coronavirus began on 27 December 2020, as in other European Union countries. The vaccination programs use four vaccines, manufactured by Pfizer/BioNTech and Moderna, which are mRNA vaccines, as well as AstraZeneca and Johnson & Johnson, which are vector vaccines. As of 12 December 2021, the total number of vaccine doses administered in Poland since the beginning of the coronavirus pandemic is over 44 million. The number of people who received both doses or who are vaccinated with a single-dose Johnson & Johnson vaccine is almost 21 million. The percentage of people vaccinated with at least one dose of the coronavirus vaccine in Poland is 55.62%, and those who are fully vaccinated constitute 54.10% of the entire population [2].

Within a year of the announcement of the pandemic by the WHO, the first mRNA vaccine, COMIRNATY (Pfizer/BioNTech), was developed [3]. The active ingredient is BNT162b2 mRNA, which encodes the full length of the SARS-CoV-2 spike protein. Lipid nanoparticle-formulated mRNA technology allows for presidential transfer of information about the immunogen to antigen-producing cells [3,4]. The mRNA vaccine functions as an antigen and an adjuvant, which, after entering the cell, stimulates a response of the immune system. TLR (Toll-like receptor 7) and MDA5 (melanoma differentiation-associated 5) mediate lymphocyte activation and effector cell diversification. In the immune response, the formation of cytotoxic T-cells and helper T-lymphocytes, as well as memory T-cells (after stimulation of the type-1 interferon), occurs. Strong Th1 cell response helps in secretion of different classes of antibodies [5,6,7]. Neutralizing and IgG antibodies were found in 96.5% and 99.9% of vaccinated patients, leading to the conclusion that the vaccine elicits a better immune response compared with COVID-19 infection (5% of patients infected with COVID-19 did not have IgG antibodies, while 85–90% developed specific antibodies) [8]. After vaccination, an instant decrease in IgA and IgM antibodies against RBD (receptor-binding domain) also occurred, which proves that these isotypes are short lived [9]. The BNT162b2 vaccine is highly effective in preventing COVID-19 and is 94–95% reliable, not only in relieving disease symptoms but also in lowering mortality [10]. In September 2021, Pfizer Inc. and BioNTech SE announced the submission of an application to the European Marketing Agency for a third (booster) dose of the vaccine to prevent infection with the virus. The application is supported by data from a phase III clinical trial involving 306 participants aged 18–55 years who received the third dose of the vaccine 4.8 to 8 months after receiving the second dose. Administration of a booster dose of BNT162b2 resulted in significantly higher titers of antibodies to neutralize the wild variant of SARS-CoV-2 compared to the levels observed at two base doses [11].

The mRNA-1273 vaccine from the pharmaceutical company Moderna is the second COVID-19 vaccine approved under the EUA in the United States and the sixth approved by the European Commission. The vaccination process consists of two doses administered intramuscularly 4 weeks apart [12]. The mRNA-1273 vaccine consists of lipid nanoparticles containing nucleoside-modified mRNAs encoding a full-length stabilized version of the SARS-CoV-2 glycoprotein trimer. The modified protein contains two proline substitutions on the S2 subunit [13]. Nanoparticles containing mRNA enter dendritic cells, which then produce and present the antigen to T lymphocytes in order to activate an adaptive immune response [5]. This is followed by stimulation of CD8^+^ and CD4^+^ T cells through the production of type I interferon [5]. The second dose of the vaccine enhances the inflammatory response through short-term changes, including in macrophages, as a result of activation of memory T cells and produced B lymphocytes [14].

The Oxford/AstraZeneca COVID-19 vaccine (codenamed AZD1222) is a replication-deficient simian adenovirus vector vaccine, which means that some essential genes were deleted and replaced by a gene coding for the spike protein. As a vector, the modified chimpanzee adenovirus ChAdOx1 was used [15]. After penetration to the cell, the transcription from DNA into mRNA begins and, in the translation, cells produce a spike protein to stimulate immune system to antibodies and memory cells against an actual SARS-CoV-2 infection creation. The body’s response process is longer than that for an mRNA vaccine, where transcription is not necessary because the vaccine already contains messenger RNA [16,17]. The vaccine has received regulatory approval in more than 100 countries, in Europe under the name Vaxzevria. It is recommended for adults over 18 years of age. Clinical studies have shown that protection begins approximately 3 weeks after the first dose and persists up to 12 weeks. Vaccinated individuals may not be fully protected until 15 days after the second dose. Likewise, as with all vaccines, vaccination with Vaxzevria may not provide protection for all vaccinated. The second dose should be administered between 4 and 12 weeks after the first dose [18].

The Janssen/Johnson & Johnson vaccine consists of a replication-incompetent recombinant adenovirus type 26 (Ad26) vector expressing the SARS-CoV-2 spike protein [19,20]. After the vector has delivered the selected sequence, it is transiently expressed, and the synthesized viral S protein elicits a response that includes the production of neutralizing antibodies as well as a cell-type immune reaction. The specific immune system response was assessed 14 and 28 days after vaccination. The Ad26.COV2.S vaccine is 66% effective in a one-dose regimen in preventing symptomatic COVID-19, with 85% efficacy in preventing severe COVID-19 and 100% efficacy in preventing hospitalization or death caused by the disease [21,22].

The efficacy of approved vaccines is dependent on viral mutations, and the approved vaccines have shown variable (reduced) efficacy against these mutant strains compared to efficacy against the original unmutated virus [23]. The factor of time since complete vaccination also affects the effectiveness of vaccines. One of the indicators of changes indicating a decrease in the effectiveness of protection against infection may be the level of post-vaccination IgG antibodies.

## 2. Materials and Methods

### 2.1. Cohort

In this work, we conducted a prospective longitudinal cohort study among employees of the Nicolaus Copernicus University in Torun, Poland. The analyses included different age categories, genders, blood groups and chronic diseases. In the context of these data, the humoral response was analyzed, the correlate of which was the presence of anti-SARS-CoV-2 IgG antibodies in the serum. The participants of the study were injected with four vaccines approved in Poland, with the most numerous group consisting of people vaccinated with Pfizer/BioNTech (58.9%) or Oxford/AstraZeneca (23.2%). Vaccinations were performed from January to August; therefore, the analyses took into account the time factor. Blood samples were collected three times: 5–6 May, 29 June and 27 September (Figure 1). In total, 954 participants were tested (Table 1). Due to the freedom to participate in the study, not all participated in the three blood donations. A total of 702 blood samples were obtained in measurement 1, 633 blood samples in measurement 2 and 722 in measurement 3. The complete set of three samples was obtained for 519 participants, and eight extreme observations were excluded from further analysis (Figure 1). In the comparative analyses of the durability of the responses, only the data from the complete collection were used. If there was a justified necessity to exclude part of the data, this was specified in the part on specific analyses.

All Nicolaus Copernicus University employees were informed by e-mail about the chance to join the studies and undergo serological tests both before receiving the first dose (in the initial stage of the research) and after receiving one or two doses of any vaccine. Employees had the possibility of three serological tests, with an interval of 2 or 3 months. Before starting the study, the participants each completed a questionnaire in which they reported age, sex, blood type, chronic diseases, vaccination dates and vaccine names, dates of COVID-19 infection (if applicable) and whether they had symptoms of the disease (if applicable). Written informed consent was obtained from all participants. The protocol and informed consent were approved by the Bioethics Commission of the Nicolaus Copernicus University in Torun at Ludwik Rydygier Collegium Medicum in Bydgoszcz, no. KB173/20.

### 2.2. Aim of Study

The aim of our study was to compare the concentrations of anti-SARS-CoV-2 IgG antibodies and to assess the relationship between antibody titers and factors such as vaccine type, previous COVID-19 infection and time of measurement, as well as age, sex and blood type. We also considered the inclusion of chronic diseases in the analysis, but, nevertheless, we did not include them due to their diversity being too large and the disproportion in numbers between the groups.

### 2.3. Enzyme-Linked Immunosorbent Assay for Detection of IgG Antibodies

Serum was centrifuged from the fresh whole blood samples. On the basis of information about the presence of SARS-CoV-2 infection and/or vaccination (type of vaccine, number of doses and dates of injection), the appropriate dilution of serum for the test was selected. A commercially available test with CE-IVD certificate (Anti-SARS-CoV-2 QuantiVac ELISA (IgG) by EUROIMMUN), which is 100% compatible with the PRNT50 neutralization assay, was used for the research [24,25]. In comparison with the microneutralization assay, Anti-SARS-CoV-2 QuantiVac ELISA showed a high level of qualitative overall agreement and a strong quantitative correlation with the results of neutralization testing [26]. The selected kit allowed the determination of the concentration of IgG antibodies with neutralizing properties against the S1 antigen of SARS-CoV-2. The assays were performed on 96-well microplates coated with the SARS-CoV-2 S1 domain (including RBD) recombinantly expressed in the human HEK293 cell line (ATCC), in the EUROIMMUN Analyzer I automated system in accordance with the recommendations of the manufacturer. We have previously described the research methodology in detail [27]. For quantification of S1-specific IgG, a 6-point calibration curve was established, and each test was run in the presence of positive and negative controls. The test results were given in standardized international units: BAU/ml (binding antibody units), by correlating the test with the new WHO reference material (NIBSC code: 20/136). Results of <25.6 BAU/ml were considered negative, ≥25.6 to <35.2 BAU/ml were borderline, and ≥35.2 BAU/ml were positive.

### 2.4. Statistical Analysis

Statistical procedures and calculations were performed with the use of the R environment (v4.1.2) and IBM SPSS Statistics (v27); all statistical tests are considered to be statistically significant if their respective *p*-values are less than 0.05.

The levels of IgG anti-SARS-CoV-2 antibodies are affected by various factors, i.e., prior COVID-19 infection, vaccine type and vaccination time, measurement time, age, gender and blood type, among many others. Since there are many factors, our data were not complete or equally representative at all factors levels, and multivariate analysis involving all factors simultaneously would be difficult to interpret and possibly not well fitted. Thus, we decided to divide the analysis into two parts.

Before the main procedures, we investigated the dependency of the time elapsed after vaccination or COVID-19 infection and of the vaccine type, since we observed that the subjects reported antibody measurements in various stages of developing immunity. In order to confirm this dependency, Pearson’s chi-squared test of independence was conducted and the effect size using Cramer’s V was computed. As a consequence of the results, time after vaccination or prior COVID-19 infection was not involved in further analysis.

Our primary aim was to analyze the concentrations of IgG anti-SARS-CoV-2 antibodies depending on three main factors: the antibody measurement phase, type of the vaccine and prior COVID-19 infection. We employed the multivariate mixed-model analysis of variance, which, however, was not the most appropriate due to not having the best model fit. Therefore, we carried out further analysis consisting of two robust two-way mixed analyses of variances [28] with use of 10% trimmed means and Huber’s Psi M-estimators, with the measurement stage as a within-subject factor and two between-subject factors: prior COVID-19 infection and the vaccine type. Since we obtained significant results, post hoc tests were applied (Games-Howell with Tukey adjustment).

Since we were particularly interested in comparisons between the IgG anti-SARS-CoV-2 antibody titers in all three stages of measurement (regardless of any other between-subject factors), a robust one-way repeated measures analysis of variance for the trimmed means with corresponding robust post hoc tests was conducted.

Next, we focused only on the between-subject factors, i.e., prior COVID-19 infection and the vaccine type. The two-way robust analysis of variance for trimmed means with interactions effects and corresponding post hoc tests were carried out.

The second part of the analysis focused on the remaining factors of interest: gender, age and blood type. Analogously, we conducted three robust two-way mixed analyses of variances with use of 10% trimmed means and Huber’s Psi M-estimators, with the measurement phase as a within-subject factor and three between-subject factors, namely, gender, age and blood type. No further post hoc tests were needed here, since we did not observe a significant impact of these factors on the IgG anti-SARS-CoV-2 antibody level.

## 3. Results

### 3.1. The Influence of Factors on the Level of Anti-SARS-CoV-2 IgG

The principal aim of this study was to compare the IgG antibody levels with respect to the within-subjects factor, which is the measurement number (1, 2, 3), also taking into account two between-subjects factors, prior COVID-19 infection (yes, no) and vaccine type (BNT162b2, mRNA-1273, ChAdOx1 nCoV-2019 and JNJ-78436735). First, since the subjects approached the measurements in various stages after having COVID-19 and vaccination, we divided the cohort (511 subjects) according to the following criteria:Group 1: up to 14 days between the planned date of the second measurement and the last dose of vaccine;Group 2: between 14 and 60 days; andGroup 3: more than 60 days.

As we suspected, Pearson’s chi-squared test of independence confirmed a significant association between the group assignment and the type of vaccine (ꭓ^2^ (6, *n* = 511) = 286.96, *p* < 0.001). The effect size for these findings, Cramer’s *V*, was moderate, 0.433. The nature of this dependency can be derived from the standardized Pearson’s residuals, which are illustrated in the association plots (Figure 1).

The association between particular group and the vaccine type is assumed only if the absolute value of standardized Pearson’s residuals is greater than 2.

This follows that group 2 mainly contains observations associated with ChAdOx1 nCoV-2019 (and an exceptionally small number of those coinciding with BNT162b2), and fewer, but still relatively many, with JNJ-78436735. Group 3 includes an over-representation of observations coinciding with BNT162b2 (and an exceptionally small number of those coinciding with ChAdOx1 nCoV-2019). Observations in the first group are allocated independently, regardless of the type of vaccine. Since such an association between the group and the vaccine type was statistically significant, we decided to carry out further analyses without taking the group into account, only considering the type of vaccine, mining the relationship between the time of vaccination (group) and the vaccine type.

In order to compare the measurements in various conditions, we decided to conduct the multivariate mixed-model analysis of variance (ANOVA) with three independent factors (within-subject): the measurement number (three levels: 1, 2, 3) and (between-subject) COVID-19 (two levels: yes, no), vaccine (four levels: BNT162b2, mRNA-1273, ChAdOx1 nCoV-2019, JNJ-78436735), with the dependent variable IgG anti-SARS-CoV-2 antibody titer. Since our data were skewed and unbalanced, before carrying out the analysis, we decided to reject incomplete (subjects not participating in all three measurements or not vaccinated) cases. Furthermore, since eight observations (antibody measurements) were extremely large (i.e., their values exceeded the upper quartiles by at least five interquartile ranges) and their credibility raised our doubts, we decided to remove them from the analysis. Finally, we analyzed *n* = 511 subjects. We illustrate the data grouped by the measurement number in Figure 2, Figure 3 and Figure 4.

IgG anti-SARS-CoV-2 antibody distributions depending on prior COVID-19 infection and vaccine type are depicted in Figure 5, Figure 6, Figure 7 and Figure 8.

Finally, data with respect to all three conditions are illustrated in Figure 9.

The multivariate mixed-model ANOVA described above was our preliminary attempt to examine the association and compare the IgG anti-SARS-CoV-2 antibody levels in all conditions, but the resulting model was not the best fitted. This was predictable due to the shapes of distributions and the amount of data at our disposal. However, we report significant results below, emphasizing that further analyses were conducted with use of the robust tests. Hence, the effect of interaction between the measurement number, prior COVID-19 infection and vaccine type occurred to be significant (Wilks’ lambda *F* (6, 1004) = 3.344, *p* < 0.001, η^2^ = 0.02). This implies that the mean values of IgG anti-SARS-CoV-2 antibodies for different stages of measurement differ significantly depending on prior COVID-19 infection and vaccine type (Figure 9). Moreover, the interaction between the measurement and prior COVID-19 infection (Figure 3) was also significant, Wilks’ lambda *F* (2, 502) = 8.408, *p* < 0.001, η^2^ = 0.032, as well as the measurement and the vaccine type (Figure 4), Wilks’ lambda *F* (6, 1004) = 5.737, *p* < 0.001, η^2^ = 0.033. The main effect of prior COVID-19 infection (Figure 5) and the main effect of the vaccine type (Figure 6) were significant (*F* (1, 503) = 51.483, *p* < 0.001, η^2^ = 0.093 and *F* (3, 503) = 41.161, *p* < 0.001, η^2^ = 0.197, respectively); however, the interaction between COVID-19 and vaccine type was not significant (*F* (3, 503) = 0.524, *p* = 0.67, η^2^ = 0.003). Further analyses were conducted with use of the WRS2 R package and consisted of robust two-way (mixed) ANOVAs using 10% trimmed means and Huber’s Psi M-estimators of location, with corresponding post hoc tests. Therefore, we report and illustrate the location parameters with trimmed means and respective confidence intervals.

### 3.2. Measurements, Prior COVID-19 Infection and Vaccine Type

In order to include the measurement number while analyzing prior COVID-19 infection and vaccine type, robust mixed ANOVAs were conducted. As we already reported, the main effects of prior COVID-19 infection (*Q* = 57.75, *p* < 0.001) and of vaccine type (*Q* = 35.32, *p* < 0.001), as well as the main effect of the measurement number (*Q* = 45.88, *p* < 0.001 for the COVID-19 model and *Q* = 30.48, *p* < 0.001 for the vaccine model), were significant (Figure 10, Figure 11 and Figure 12 and Table 2).

The sample characteristics of the IgG anti-SARS-CoV-2 antibody titers with respect to all considered factors are presented in Table 2.

Notably, the interactions between the measurement number and prior COVID-19 infection (*Q* = 10.57, *p* < 0.001) and between the measurement number and vaccine type (*Q* = 18.60, *p* < 0.001) were also significant (Figure 13 and Figure 14 and Table 2).

Since the post hoc tests in mixed ANOVA in WRS2 package do not give the clear interpretation of the differences on all factors’ levels, we applied further analyses while comparing the IgG anti-SARS-CoV-2 antibody titers. To break the interaction effects, we conducted pairwise comparisons for each of three measurements for both between-subject factors with use of the Games-Howell tests with Tukey’s adjustment for multiple comparisons.

#### 3.2.1. Measurements and Prior COVID-19 Infection

Post hoc analysis confirmed that in all measurements, the subjects without prior COVID-19 infection had significantly lower (*p* = 0.008, <0.001, <0.001, respectively) anti-SARS-CoV-2 IgG antibody titers than those with prior COVID-19 infection (Figure 3 and Figure 13 and Table 2).

#### 3.2.2. Association of Measurements and Vaccine Type

Comparing IgG anti-SARS-CoV-2 antibody titers between the measurements and vaccine type, we conclude that there are significant differences between all types of vaccines in all measurements (*p* < 0.007), except those mentioned below:In the first measurement, there are no significant differences between BNT162b2 and mRNA-1273 (*p* = 0.263), and between ChAdOx1 nCoV-2019 and JNJ-78436735 (*p* = 0.076).In the second measurement, there are no significant differences between BNT162b2 and JNJ-78436735 (*p* = 0.996), and between ChAdOx1 nCoV-2019 and JNJ-78436735 (*p* = 0.856).In the third measurement, there are no significant differences between BNT162b2 and JNJ-78436735 (*p* = 0.966), between ChAdOx1 nCoV-2019 and JNJ-78436735 (*p* = 0.665) and between JNJ-78436735 and mRNA-1273 (*p* = 0.220).

This can be compared in Figure 4 and Figure 14 and Table 2.

#### 3.2.3. Measurements

We found it particularly interesting to investigate the differences between all three measurements, which, in the preliminary multivariate mixed-model ANOVA, were significant (Wilks’ lambda *F* (2, 502) = 37.227, *p* < 0.001, η^2^ = 0.129). The robust repeated-measures ANOVA for the trimmed means comparing IgG anti-SARS-CoV-2 antibody titers proved significant differences (*F* (1.48, 602.04) = 64.595) between the measurement levels. Corresponding post hoc tests resulted in the following conclusions: the first and the second measurements of IgG anti-SARS-CoV-2 antibody titers do not differ (*p* = 0.05), and they are significantly higher than the last measurement (*p* = 0.025 and 0.017, respectively). This can be compared with Figure 12 and Table 2.

#### 3.2.4. Association of Prior COVID-19 Infection and Vaccine Type

Next, we focused only on the between-subject factors, regardless of the measurement number. The results of robust two-way ANOVA based on trimmed means confirmed no significance of the interaction effect, which implies that there are no significant differences in IgG anti-SARS-CoV-2 antibody titers (*p* = 0.172) between prior COVID-19 infection levels and the vaccine types (regardless of the measurement type). The results also proved significant differences in prior COVID-19 infection (*p* = 0.001) and the vaccine type (*p* = 0.001). Furthermore, we applied corresponding post hoc tests for M-estimators of location using Huber’s Psi. As reported above, the IgG anti-SARS-CoV-2 antibody titers for the subjects without prior COVID-19 infection were significantly lower than for COVID-19-recovered subjects (*p* < 0.001), regardless of the measurement (Figure 10). Comparing the vaccine types without taking the measurement number into account, we conclude that the significant differences in mean IgG anti-SARS-CoV-2 antibody titers are observed between all types of vaccines (*p* < 0.015), except for ChAdOx1 nCoV-2019 vs. JNJ-78436735 (*p* = 0.944). The highest IgG anti-SARS-CoV-2 antibody titers were detected for mRNA-1273. Significantly lower antibody levels were observed for BNT162b2, and the lowest and similar values correspond to ChAdOx1 nCoV-2019 and JNJ-78436735 (Figure 11).

### 3.3. Gender, Age Category and Blood Group

Eventually, we decided to look at the other factors that might have influenced the IgG anti-SARS-CoV-2 antibody titer: gender (male, female), age category (< 30, 30–40, 40–50, 50–65, 65+), and the AB0 blood groups (0, A, B, AB). Similar to the case of prior COVID-19 infection and vaccine type, robust mixed ANOVAs were conducted, one model for each of the three mentioned between-subject factors and within-subject measurement number. The analysis considering the blood type was restricted only to the subjects who declared a known blood group (*n* = 420). It occurred that none of the interactions were significant: neither the measurement number and gender (*Q* = 0.569, *p* = 0.567), nor the measurement number and age category (*Q* = 1.057, *p* =0.403) and measurement and blood group (*Q* = 0.418, *p* = 0.910). Respective box plots are presented in Figure 15, Figure 16 and Figure 17.

None of the main effects were significant—gender (*Q* = 0.983, *p* = 0.322), age category (*Q* = 0.443, *p* = 0.777) and blood group (*Q* = 0.488, *p* = 0.745)—except the measurement number, as reported in the previous paragraph. This means that in the view of our analysis, neither gender, nor age, nor blood type have any influence on IgG anti-SARS-CoV-2 antibody titers (Figure 18, Figure 19 and Figure 20).

## 4. Discussion

The aim of the study was to compare four vaccines used in Poland for one of the parameters of the humoral response, the concentration of IgG antibodies against S protein. The analyses included the time factor, previous COVID-19 infection and parameters such as age, gender and blood type of participants. In comparative analyses, we noticed significant differences in the levels of post-vaccination antibodies, depending mainly on the type of vaccine, the time elapsed since full vaccination, and previous SARS-CoV-2 infection. First, we noted that mRNA vaccines stimulated more intensive IgG antibody production compared to vector vaccines. In our study, the highest levels of antibodies, regardless of the time of measurement, were obtained in serum samples obtained from people vaccinated with Moderna. For mRNA vaccines, significant advantages over conventional vectored vaccines were demonstrated in terms of efficacy, safety and a broad spectrum of activation of immune response components [29,30]. This advantage is a consequence of, among others factors, the higher immunogenicity due to improved translation efficiency [31], lower risk of insertion-induced mutagenesis [32], ability to produce higher levels of neutralizing antibodies by activating CD4^+^ and CD8^+^ T cells at relatively low doses [31,33] and ease of design and possible modification of the structure in relation to new virus variants [34]. Scientists have proven that the antibody levels are several times higher after mRNA vaccination than vector vaccination in the first few weeks. For Pfizer, antibody levels reduced from a median of 7506 U/mL at 21–41 days to 3320 U/mL at 70 or more days. For the AstraZeneca vaccine, antibody levels reduced from a median of 1201 U/mL at 0–20 days to 190 U/mL at 70 or more days [35]. According to the information in the leaflet, the efficacy range of the AstraZeneca vaccine is close to the efficacy of Johnson & Johnson, but lower than Pfizer and Moderna. Steensels et al. [36] compared antibody levels between mRNA vaccines. Like us, they observed a more intense immune response with the Moderna vaccine. The results of their studies show that in participants vaccinated with two doses of mRNA-1273, the mean titer of neutralizing antibodies was 3836 U/ml, and in those vaccinated with BNT162b2, the mean titer of neutralizing antibodies was 1444 U/ml. As a possible difference in the immunogenicity of the compared mRNA vaccines, they indicated higher mRNA content in mRNA-1273 compared with BNT162b2 and a longer interval between the priming and boosting doses for the Moderna vaccine (4 weeks vs. 3 weeks for Pfizer/BioNTech vaccine) [36].

Vaccine effectiveness is generally expected to slowly decrease over time. The study published by Cohn et al. [37] shows that just three months after full vaccination with Pfizer/BioNTech, Moderna or Johnson & Johnson, the effectiveness of the vaccines begins to significantly and disproportionately decrease. The slowest rate of decline in protection was noticed in the case of Moderna, and it was quite intensified in people vaccinated with the single-dose Johnson & Johnson vaccine. At the end of September, six months after vaccination, protection for patients who received Moderna was 58% (in March, it was 89%). In the case of Pfizer/BioNTech, there was a decrease from 87% to 45%. As for Johnson & Johnson, the level dropped from 86% to just 13% in six months. This means that just six months after full vaccination with this vaccine, it practically no longer protects against coronavirus infection. However, the authors emphasize that all vaccines after six months maintain a high protective barrier against death as a result of COVID-19 infection. In the case of Moderna, the risk of death was reduced by 76% compared to the unvaccinated. For Pfizer/BioNTech, the risk of death was reduced by 70%, and for J&J, it was reduced by 52% [37].

The most frequently represented group in our study were people vaccinated with Pfizer/BioNTech. Most people reporting for the first blood collection were fully vaccinated. The level of antibodies in these people was significantly higher than in those who had been injected with the vectored vaccine, and lower than those who received the Moderna vaccine. From our previous study [27], we know that the highest level of antibodies to this vaccine is determined in the 2–3 weeks after the second dose, and after this period, it gradually slows down, and the pace of these changes is most intense until 4 months after full vaccination. Eight months after complete vaccination, 6 subjects tested positive for antibodies and 46 were negative (<100 BAU/mL). In our BNT162b2-vaccinated cohort, only three people obtained positive antibodies for SARS-CoV-2 after full vaccination. Studies conducted by Naaber at al. [38] showed elevated levels of IgG S-RBD antibodies three weeks after the first dose of vaccination. Studies have shown a significant increase in IgG antibodies after the second dose of the vaccine. However, 12 weeks after the second dose, there was a significant decrease in antibodies. A decrease in antibody levels between the first and sixth weeks after the second dose of the vaccine was present in the entire study group and decreased by an average of 45% between these two time points. After six months, IgG S-RBD levels were only 2–25% of the peak level of antibodies, detected one week after the second dose [38]. It was also shown that in people who were not infected with SARS-CoV-2, after administration of the Pfizer/BioNTech mRNA vaccine, antibody levels were initially higher, but also decreased faster compared to people who had the disease. Six months after vaccination, antibody levels dropped to an average of 7% of their peak level, which is comparable to antibody levels in patients who recovered from COVID-19 [38,39]. Studies have shown that a longer interval between two doses of a vaccine promotes a better immune response, as shown with the Pfizer/BioNTech vaccine. Data suggest that two-dose vaccination may induce a better immune response to SARS-CoV-2 than homologous single-dose vaccination [40].

We observed the highest levels of antibodies in each measurement in serum from people vaccinated with Moderna. At 8 months after vaccination, none of the participants had obtained a low antibody titer, and only one participant had tested positive for the presence of coronavirus. The phase III clinical trials results indicate that the effectiveness of the Moderna COVID-19 vaccine after two doses was 94.1% among people not previously infected with the SARS-CoV-2 virus [41]. In a study by Widge et al. [42], after a 100 μg dose of the mRNA-1273 vaccine, high levels of binding and neutralizing antibodies were produced, which decreased slightly over the course of the study but remained elevated in all participants 3 months after booster vaccination. Vaccine mRNA-1273 is characterized by high peak responses that decline rapidly after 6 months. A further decline was also recorded after 8 months. In general, antibody titers in mRNA-1273 vaccine recipients were higher than in BNT162b2 vaccine recipients. Anderson et al. [13] conducted studies in which a vaccine from Moderna induced high levels of binding and neutralizing antibodies in the elderly. Additionally, the study that correlated with the timing and dose of the vaccine was similar in both younger and older subjects. Antibody levels in patients who received a booster dose were similar to those seen in people who are recovering from mild to severe disease [13]. Preclinical and early research results suggest that the mRNA-1273 vaccine caused an IgG antibody response against the S protein and specific neutralizing antibodies to SARS-CoV-2 virus for several months after vaccination. In another study, Pegu et al. [43] found that mRNA-1273-induced antibody activity against SARS-CoV-2 variants was shown to be sustained 6 months after the second dose, albeit at a lower level than the peak activity. Moreover, the binding antibodies recognizing SARS-CoV-2 variants B.1.351 and B.1.617.2 remained high [43].

People vaccinated with Oxford/AstraZeneca were our second-largest group. Due to the implementation of the vaccination program, the majority of study participants had injections at a similar time. In the first stage, these subjects were measured after the first dose, and in the second stage, at the peak after the second dose (6–8 weeks after the second injection). This vaccine induced weaker production of anti-SARS-CoV-2 antibodies compared to mRNA vaccines, although the antibody levels observed over time were more stable. After the first dose, up to 46% of participants obtained a low-positive result (<100 BAU/mL), and 19 persons were negative for the presence of vaccine antibodies. Three months after the second dose, the negative result was confirmed in four patients, and the slow-positive result was found in twenty-seven patients. Only three people had SARS-CoV-2 infection after full vaccination. The results of studies suggest that a single dose of a vaccine induced high levels of antibodies to the RBD (receptor-binding domain) and ACE2-blocking (angiotensin I converting enzyme 2) antibodies, which was greater than immune responses in patients who experience a mild or asymptomatic natural infection. The T cell responses were comparable to natural infection. In those who previously had COVID-19, a single dose induced very high levels of ACE2-blocking antibodies and antibodies to RBDs [44]. Mishra et al. [45] analyzed the kinetics of the change in antibody levels for this vaccine and showed a significant decrease in antibody levels 6 months after injection. In their research, as in ours, the history of prior SARS-CoV-2 infection was the only significant factor in antibody levels for ChAdOx1-2019. Some countries are considering giving patients a third booster dose. At the same time, the temporary discontinuity in the vaccines’ supply suggests considering the effectiveness of immunity response depending on the interval between doses. It was observed that antibody titers were higher in patients with a longer interval between their first and second doses. Antibody titers were significantly higher following a third dose when compared with the response 28 days after a second dose. A longer period of time between doses leads to increased antibody titers after the second shot [46].

Depending on the predominant SARS-CoV-2 variant in a given period and the time elapsed since the last injection, the effectiveness of vaccines is updated in terms of protection against infection. The presence of IgG antibodies against SARS-CoV-2 is one of the indicators that can be analyzed, which indirectly shows the status of the immune system in relation to potential infection. It is also a marker that is readily available in routine diagnostics. In our previous study, we showed that the highest levels of anti-spike immunoglobulins G are reached 2–3 weeks after the second dose of the Pfizer/BioNTech vaccine [27]. For vector vaccines, the peak of the immune response is about 6 weeks after full vaccination. Over time, the concentration of anti-SARS-CoV-2 IgG antibodies naturally declines, with the rate of antibody loss being higher in those vaccinated with mRNA preparations (especially those who are seronegative prior to the first injection). The rate of antibody loss is greatest in the first 4 months after taking the second dose and is approximately 50% every 4 weeks. In the fourth to sixth month after full vaccination, the antibody concentration is below the titer before taking the second dose. Although the decline in antibody titers is slower around 6–8 months, it is below 10%. Nearly 2% of our research participants during this period were negative for the presence of anti-SARS-CoV-2 IgG antibodies. Vector vaccines, and, especially, Vaxzevria, analyzed in detail in this study, are more stable in terms of the loss of anti-SARS-CoV-2 IgG antibodies. Studies conducted in the U.K. by Pouwels et al. [47] showed that the effectiveness of the Pfizer/BioNTech vaccine declines faster than others for the coronavirus mutation with a high viral load (such as the Delta variant). According to the manufacturer’s specification, the Pfizer/BioNTech vaccine is approximately 90% effective against infections with high viral load, but within a month after the second dose. However, this effectiveness drops to 85% after two months and 78% after three months. According to these data, the protective properties of this vaccine are lost by several percentage points. By comparison, the effectiveness of the Oxford/AstraZeneca vaccine decreased by only six percentage points (the equivalent protection was 67%, 65% and 61% over the same period). Interestingly, a single dose of the Moderna vaccine shows similar or greater efficacy against the Delta variant as single doses of other vaccines [47,48]. Our findings and the available results of studies by other scientific groups show that the vaccination story of the Oxford/AstraZeneca vaccine begins with a much lower initial effectiveness (as well as several times lower antibody levels compared to mRNA vaccines). However, it has been calculated that approximately 5 months after taking the second dose, both vaccines may have more of the same effectiveness in preventing infections with any variant, including Delta. Despite the observed declines in effectiveness in preventing infection, both vaccines should demonstrate a high level of protection in preventing hospitalization or death from COVID-19 [47,49,50]. Collier et al. [51] described in detail the kinetics of changes in the immune response for BNT162b2, RNA-1273 and Ad26.COV2.S, including live-virus neutralizing antibody titer, a pseudovirus neutralizing antibody titer and a binding antibody titer against RBD. They observed a several-fold reduction in median titers 8 months after vaccination. As noted, all three vaccines mentioned during this period displayed similar median pseudovirus neutralizing antibody titers against the SARS-CoV-2 B.1.617.2 variant. Collier et al. [51] also analyzed the kinetics of the response between the three vaccines. They noticed that the compared peak antibody titers were the highest for mRNA-1273. A slightly less intense humoral response has been observed with BNT162b2. These two mRNA vaccines lost antibodies rapidly in the first 6 months after the second dose. The Ad26.COV2.S vaccine elicited lower initial antibody responses, but the antibody concentrations obtained were relatively stable throughout the 8-month follow-up period. The immunodeficiency trends based on the selected marker antibodies are consistent with our observations, with the difference that the most intense reduction in IgG antibodies against the S1 subunit from our study took place up to 4 months after the administration of the second dose of mRNA vaccines. Six months after vaccination, patients who received the Moderna vaccine had protection against infection of 58%, with the Pfizer vaccine dropping to 45%. With Johnson & Johnson, the protection level was just 13%. After six months, however, all vaccines remain highly effective in protecting against death from COVID-19 infection. Moderna’s risk of dying from coronavirus is 76% lower, Pfizer’s is 70% lower and J&J’s drops to 52% [37].

One of the key aspects of the analysis of this study was the search for the factors determining the intensity of the humoral response. Therefore, we analyzed the potential influence of age, gender and blood group on anti-S antibody levels. Contrary to our assumptions, we did not observe any significant influence of any of them on the immunoglobulin G titer, and the only factor significantly influencing the level of anti-SARS-CoV-2 IgG was prior natural immunization as a result of virus infection. It seemed that the age of the patient is the major factor influencing the humoral response to vaccines [52,53]. The values of anti-SARS-CoV-2 antibodies after vaccination are higher compared to the elderly [54]. This applies to all age categories studied. These differences are observed even in young patients in the age range of 12–25 years [52]. However, these observations are more pronounced in the later age categories. Elderly people are also significantly more likely to have poor or no post-vaccination humoral response [54]. The reason for this is a decrease in the functionality of the immune system, inflamm-aging and immunosenescence. An additional factor influencing this response is the existence of comorbidities that also develop in proportion to the age of the person [55,56]. Gender is another factor that may influence the humoral response after vaccination. Much of the literature data indicate higher levels of anti-SARS-CoV-2 IgG in women compared to men [53,57]. Genetic factors may play a role here. There are more genes (including miRNAs) on the X chromosome that are responsible for modulating the immune response [57,58]. Moreover, women of childbearing age are characterized by a stronger overall immune, inflammatory, antiviral and humoral immune response. This is due to the effects of estrogens, which contribute to the enhanced response to both infection and vaccination. On the other hand, in men, testosterone suppresses the immune response after vaccination by affecting androgen receptors and immune cells. As is known, the effect of these hormones is canceled in elderly women and men [55,57]. It was surprising to us that there was no correlation between antibody concentration, gender and age. The last of the factors we analyzed that may have a potential impact on the vaccine immune response was the blood group. Similarly to the other factors, we did not find a significant correlation between the ABO type and the level of immunoglobulin G against the protein S. Scientists have noticed that many more people with blood groups A and AB had more antibodies compared to people with blood group O. The difference was 2.5% between these groups. Blood group A may increase the risk of developing COVID-19 due to the “attraction” of the SARS-2 coronavirus and, more particularly, the receptor binding domain (RBD) located at the top of the spike protein (S1 subunit) in blood group A antigens, which are found on the cells of the airways—that is, the RBD of the new coronavirus joins the ACE2 receptors on the surface of the airway cells, which allows the development of infection [59,60].

## 5. Conclusions

Successively published studies in the field of vaccine effectiveness analysis confirm vaccines’ high effectiveness in protection against COVID-19. The effectiveness of vaccines varies not only in terms of variants but also over time. The factor that has the greatest impact on the intensity of the vaccine response (as well as indirectly its durability), regardless of its type, is previous SARS-CoV-2 infection. Thanks to studies carried out in vaccinated people, we know that the effectiveness of vaccines decreases over time after full vaccination. Therefore, some countries have decided to administer another dose of the COVID-19 vaccine, which seems to be the right decision.

## Data Availability

The data presented in this study are available on request from the corresponding author. The data are not publicly available due to their containing information that could compromise the privacy of research participants.

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
