# Peer review of "Anti-SARS-CoV-2 IgG against the S Protein: A Comparison of BNT162b2, mRNA-1273, ChAdOx1 nCoV-2019 and Ad26.COV2.S Vaccines"

_vaccines, 2022, doi:10.3390/vaccines10010099_

Round 1
Reviewer 1 Report
The proposed scientific article is of interest to epidemiologists, virologists and clinical infectious disease specialists. The authors publish interesting data on humoral immunity, which is formed after the use of various vaccines against COVID19. I think that the article can be published.
Author Response
Thanks a lot for so much meaningful review for us. We are very pleased that the effort of our work has been appreciated.
Reviewer 2 Report
The manuscript reports the findings from a study of seroprevalence after the vaccination against COVID-19 in a cohort of Polish individuals. The authors evaluated the estimates of IgG dosage at three subsequent measurements.
Overall, the manuscript is of interest to the broad readership – findings are of clinical interest and might add pieces of evidence to the field.
My main concern refers to the length of the manuscript and sharpness of the methodology. The aim of the study is not clear, unless the whole manuscript is read – and, unfortunately, reading is hampered by the length of the manuscript and the erratic use of the English language.
Some paragraphs do not seem to be strictly pertinent [e.g.: Introduction, last paragraph], and the Discussion is definitely too verbose; some phrases throughout the manuscript remain obscure (by the way, ‘analyzes’ should be replaced with ‘analyses’).
From the methodological standpoint, I observe that the primary endpoint is not clearly stated. What is the main hypothesis under evaluation? Are there secondary endpoints? In fact, the methods section does not describe the statistical approach – what about models, variables, tests, corrections, power, etc.? – pieces of information are provided in the Results section, thus making reading even more difficult. The lack of accurate description of the statistical approach hampers to interpret the data reported some instances [e.g.: figure 1].
Moreover, the following comments could be addressed to improve the scope of the article.
- In the introduction [1st paragraph] the current COVID-19 figures in Poland are provided – however, the landscape is rapidly evolving, the reference period should be specified, as well as a dynamic source of data.
- The workflow of enrolment and sample analyses should be summarized, graphically or in the text at present it is not clear how 515 participants were selected (by the way, ‘…4 extreme observations were excluded…’ [page 4 line161] is rather obscure).
- As the study comprised longitudinal measurements, it should be clear to the reader whether and how both the vaccination regimen and the vaccinated people changed over time.
- The conclusions, as stated in the last paragraph [page 23 lines 663-668] are rather bold and, most important, are not based on the findings reported in the manuscript.
Author Response
The manuscript reports the findings from a study of seroprevalence after the vaccination against COVID-19 in a cohort of Polish individuals. The authors evaluated the estimates of IgG dosage at three subsequent measurements.
Overall, the manuscript is of interest to the broad readership – findings are of clinical interest and might add pieces of evidence to the field.
Thank you very much for your opinion so much for us. We are very pleased that the effort of our work has been appreciated.
My main concern refers to the length of the manuscript and sharpness of the methodology. The aim of the study is not clear, unless the whole manuscript is read – and, unfortunately, reading is hampered by the length of the manuscript and the erratic use of the English language.
The manuscript has been improved in line with the guidelines - substantive, technical and linguistic changes have been made. We added an appropriate comments concerning the factors analysed in the manuscript.
We removed from the text of the Introduction and Discussion information about the effectiveness of vaccines in relation to SARS-CoV-2 variants and the impact of vaccines on the cellular response in order to improve the readability of the manuscript. We also corrected the text in terms of stylistic and linguistic errors. We have performed professional linguistic proofreading at MDPI English Editing Services.
Some paragraphs do not seem to be strictly pertinent [e.g.: Introduction, last paragraph], and the Discussion is definitely too verbose; some phrases throughout the manuscript remain obscure (by the way, ‘analyzes’ should be replaced with ‘analyses’).
The manuscript has been redrafted and professional linguistic corrections have been made.
From the methodological standpoint, I observe that the primary endpoint is not clearly stated. What is the main hypothesis under evaluation? Are there secondary endpoints? In fact, the methods section does not describe the statistical approach – what about models, variables, tests, corrections, power, etc.? – pieces of information are provided in the Results section, thus making reading even more difficult. The lack of accurate description of the statistical approach hampers to interpret the data reported some instances [e.g.: figure 1].
Thank you for this valuable remark, which helped us to improve the manuscript. We added an appropriate paragraph concerning the models and variables analysed in the manuscript in the Statistical analysis section. Previously these information were indeed presented in the Results section, and are sequentially described, according to the chronology of the data analysis. All variables (referred to as between- and within-subject factors, i.e. measurement number, COVID-19 prior infection, age category, gender, vaccine type, blood type), tests (chi-square test of independence in 3.1),, models (in ANOVA with described design and robust ANOVAS in 3.1, 3.2, 3.2.3, 3.2.4, 3.3), corrections (Tukey's in 3.2), effect sizes (eta-squared in 3.1), post-hoc tests (Games – Howell in 3.2), statistics (F, Q, chi-squared in 3.1-3.3), etc. were reported and mentioned in subsections 3.1 – 3.3. Since there were many tests conducted during the data analysis, we tried to describe them as reliable as possible in appropriate subsections of Section 3. Additionally, we added the degrees of freedom to the Q statistics reported in Section 3. We hope that introducing these information in the Statistical analysis section will positively affect the clarity of the publication and meets the Reviewer's requirements.
Moreover, the following comments could be addressed to improve the scope of the article.
- In the introduction [1st paragraph] the current COVID-19 figures in Poland are provided – however, the landscape is rapidly evolving, the reference period should be specified, as well as a dynamic source of data.
We have added an update of the data, including the period it concerns.
- The workflow of enrolment and sample analyses should be summarized, graphically or in the text at present it is not clear how 515 participants were selected (by the way, ‘…4 extreme observations were excluded…’ [page 4 line161] is rather obscure).
In order to compare the antibody titer in all three conditions (measurement numbers), we had to omit uncomplete data in the main analysis, this is why we had 511 complete cases. We clarified this in a paragraph on p.6 line 262-268:
Was: „Since our data was skewed and unbalanced, before the analysis was carried out, we decided to reject incomplete (subjects not participating in all 3 measurements or not vaccinated) and few extreme observations. Finally, we analysed n=511 subjects. We illustrate the data grouped by the measurement number on Figure 2, 3 and 4.”
Is: Since our data was skewed and unbalanced, before the analysis was carried out, we decided to reject incomplete (subjects not participating in all 3 measurements or not vaccinated) cases. Further, since eight observations (antibody measurements) were extremely large (i.e. their values exceeded the upper quartiles by at least five interquartile ranges) and their credibility raised our doubts, we decided to remove them from analysis. Finally, we analysed n=511 subjects. We illustrate the data grouped by the measurement number on Figure 2, 3 and 4.
We have also prepared a graphic workflow (Diagram 1) in the section on the studied population.
- As the study comprised longitudinal measurements, it should be clear to the reader whether and how both the vaccination regimen and the vaccinated people changed over time.
The manuscript describes how to calculate the post-vaccination time for participants. This data is also marked on the newly prepared diagram.
- The conclusions, as stated in the last paragraph [page 23 lines 663-668] are rather bold and, most important, are not based on the findings reported in the manuscript.
This excerpt provides data on the results of other research groups (hence the citations; we do not suggest that this is a conclusion based on our research), in which antibody levels correlated with factors such as gender, blood type or age. Data on the influence of these factors on the vaccine response are inconsistent. In our study, we did not obtain a correlation between the level of IgG antibody against the S protein and the factors mentioned, however, they are described in the scientific literature and we wanted to describe it as well. In both the Results and the Discussions, we clearly emphasize that, unfortunately, we did not obtain such correlations in our study.
On behalf of the co-authors, thank you for the time to prepare the Review and all valuable comments that improved our manuscript.
Reviewer 3 Report
Authors compared 4 vaccines used in Poland and showed significant differences in the levels of post-vaccination antibodies, depending mainly on the type of vaccine, the time elapsed since full vaccination, as well as previous SARS-CoV-2 infection.
- Please discuss possible cause that the level of antibodies in people vaccinated with Pfizer/BioNTech was significantly higher than in those who had been injected with the vectored vaccine and lower than that of the Moderna.
- The level of antibodies after vaccination are affected by various factors, such as age, gender, comorbidities, and type of vaccine. Please show the data of multivariate analysis to emphasize the results.
- Please exhibit the details of chronic disease in Table 2.
- There are spelling mistakes, line 165,472 etc. These should be corrected.
Author Response
Authors compared 4 vaccines used in Poland and showed significant differences in the levels of post-vaccination antibodies, depending mainly on the type of vaccine, the time elapsed since full vaccination, as well as previous SARS-CoV-2 infection.
Please discuss possible cause that the level of antibodies in people vaccinated with Pfizer/BioNTech was significantly higher than in those who had been injected with the vectored vaccine and lower than that of the Moderna.
In the first paragraph of the Discussion, we attempted to explain the possible reasons for the differences in the intensity of the vaccine response between mRNA and vector vaccines. As recommended, we have added information on the possible reasons for the observed differences between the Moderna and Phizer/BioNTech preparations. “Steensels et al. [36] compared antibody levels between mRNA vaccines. Like us, they observed a more intense immune response with the Moderna vaccine. The results of their studies show that in participants vaccinated with 2 doses of mRNA-1273, the mean titre of neutralizing antibodies was 3836 U/ml, and in those vaccinated with BNT162b2 1444 U/ml. As a possible difference in the immunogenicity of the compared mRNA vaccines, they indicated the higher mRNA content in mRNA-1273 compared with BNT162b2 and the longer interval between priming and boosting dose for Moderna vaccine (4 weeks vs 3 weeks for Pfizer/BioNTech vaccine) [36].”
The level of antibodies after vaccination are affected by various factors, such as age, gender, comorbidities, and type of vaccine. Please show the data of multivariate analysis to emphasize the results.
Thank you, this is a valuable remark.We added an appropriate paragraph concerning the models and factors analysed in the manuscript in the Statistical analysis section. Since there are many factors involved in analysis (some of them insignificant), we decided to present data on several graphs to make it more legible (instead of one possibly difficult to interpret). We belive that what is mentioned by the Reviewer 2 is already presented in Fig. 9, 18, 19 and 20 and meets the Reviewer's requirements.
Please exhibit the details of chronic disease in Table 2.
The participants of the study completed a questionnaire in which they declared the presence of diseases. We considered the inclusion of chronic diseases in the analyses, but nevertheless, the analysis did not include them due to their too large diversity and disproportion in numbers between the groups.
Due to the fact that in the further analysis we decided not to present this data, we decided to omit them also in Table 2 (now Table 1), in order not to introduce information "noise", by including data for which there are no analysis results. We have included an additional comment in the text explaining the omission of this factor.
There are spelling mistakes, line 165,472 etc. These should be corrected.
The manuscript has been improved in line with the guidelines - substantive, technical and linguistic changes have been made.
On behalf of the co-authors, thank you for the time to prepare the Review and all valuable comments that improved our manuscript.
Reviewer 4 Report
The work compares the level of anti-protein S antibodies in a population of 954 individuals (convalescent, COVID-19 non-patients) vaccinated with three different vaccines by immunoassay. Furthermore, it statistically compares the relationship between vaccines, gender, age, and blood groups of participants.
The manuscript is clear and reasonable well written, and the prospective longitudinal study involves many individuals in the cohort. Properly employed Statistical methodologies. Tables and figures are necessary. However, a better resolution of figures would be adequate.
Affiliations must follow the procedure standards. (1) References need standardization: many are cited as preprint without doi, and the reader cannot access them; others are written in lowercase while others in uppercase; some with volume and number, others without number; scores, ...(2) Must include Ref 10 in the middle of the text
Author Response
The work compares the level of anti-protein S antibodies in a population of 954 individuals (convalescent, COVID-19 non-patients) vaccinated with three different vaccines by immunoassay. Furthermore, it statistically compares the relationship between vaccines, gender, age, and blood groups of participants.
The manuscript is clear and reasonable well written, and the prospective longitudinal study involves many individuals in the cohort. Properly employed Statistical methodologies. Tables and figures are necessary. However, a better resolution of figures would be adequate.
Thank you very much for your opinion so much for us. We are very pleased that the effort of our work has been appreciated. The manuscript has been improved in line with the guidelines - substantive, technical and linguistic changes have been made. All figure files are saved in the tiff format in 700 DPI (in the submissions we uploaded them in a .zip file), which should be adequate (perhaps pasting the image file into a Word file reduces the resolution).
Affiliations must follow the procedure standards. (1) References need standardization: many are cited as preprint without doi, and the reader cannot access them; others are written in lowercase while others in uppercase; some with volume and number, others without number; scores, ...(2) Must include Ref 10 in the middle of the text
We updated the list of publications using EndNote.
On behalf of the co-authors, thank you for the time to prepare the Review and all valuable comments that improved our manuscript.
Round 2
Reviewer 2 Report
The manuscript is the revised version of an article reporting a longitudinal study on humoral response to ant-COVID-19 vaccines as ascertained in a cohort of Polish participants. Antibody concentrations were measured at three time points, from May to Sept 2021; vaccines had been administered from Jan to Aug 2021.
Comparing the revised version with the previous one, it appears that the Authors partly solved the issues raised by the reviewer in the previous review process. Namely, the methodology still suffers from some weaknesses: the number of diagrams and text lines do not help to catch the main message.
As far as I understood, the main objective of the work was to establish whether the antibody response to SARS-CoV-2 depends on time from vaccination, type of vaccination, and previous SARS-CoV2 infection. However, it is not clear which is the primary objective of the study, whether the effect of time lapse or vaccine type or previous infection.
It is well known that the level of humoral immunity is a function of time elapsed from natural infection, if occurred, and from vaccination. Recent literature describes the time-dependent dynamics of antibody titers after vaccination ant-COVID-19.
As far as concerns the decay of antibody titer over time, since the three measurements were on the same date for all study participants, T0 – i.e. the date of vaccination – is fundamental to estimate the time variable. Yet, the distribution of this variable was not described.
I noticed that the previous reviewer had recommended to describe the study workflow. The revised version includes a table describing the allocation of participants, but not the distribution of the time elapsed from vaccine administration, as well as from natural infection.
The Authors divided the time variable in 3 groups (based on what considerations, since the distribution was not examined?) and found that the vaccine types were not evenly distributed across groups – see lines 235-238. Then, quite surprisingly, they declared that the variable was not taken into account just because it was significantly associated - see lines 198-204 and 254-257: ‘Since such an association between the group and the vaccine type was statistically significant, we decided to carry out further analyses without taking the group into account, only considering the type of vaccine, mining the relationship between the time of vaccination (group) and the vaccine type’. Based on their own declaration, thus, the results may be heavily biased, because a significant predictor was not considered in the analyses.
By the way, in other instances the authors state that the methods employed have been changed during the course of the analyses, depending on findings – see for instance lines164-166, 194-197, 207-210. This looks quite unusual.
Author Response
The manuscript is the revised version of an article reporting a longitudinal study on humoral response to ant-COVID-19 vaccines as ascertained in a cohort of Polish participants. Antibody concentrations were measured at three time points, from May to Sept 2021; vaccines had been administered from Jan to Aug 2021.
Comparing the revised version with the previous one, it appears that the Authors partly solved the issues raised by the reviewer in the previous review process. Namely, the methodology still suffers from some weaknesses: the number of diagrams and text lines do not help to catch the main message.
In our opinion the removal of some of figures or other parts of the manuscript would impoverish the manuscript. We wanted a multi-faceted presentation of the data discussed in the manuscript. Figures are an integral part of descriptions and, in our opinion, throwing out some of them will make interpretation difficult. We would be grateful to the Reviewer for pointing out particular elements that should be deleted or shortened.
As far as I understood, the main objective of the work was to establish whether the antibody response to SARS-CoV-2 depends on time from vaccination, type of vaccination, and previous SARS-CoV2 infection. However, it is not clear which is the primary objective of the study, whether the effect of time lapse or vaccine type or previous infection.
The main objective of work, as stated in lines 203-205, was to establish whether the antibody response to SARS-CoV-2 depends on the antibody measurement phase (not the time from vaccination), type of vaccination and previous SARS-CoV-2 infection.
It is well known that the level of humoral immunity is a function of time elapsed from natural infection, if occurred, and from vaccination. Recent literature describes the time-dependent dynamics of antibody titers after vaccination ant-COVID-19.
Yes, this is what we also note (see lines 196-202).
As far as concerns the decay of antibody titer over time, since the three measurements were on the same date for all study participants, T0 – i.e. the date of vaccination – is fundamental to estimate the time variable. Yet, the distribution of this variable was not described.
Unfortunately, one of the study limitations was an unfavourable situation of subjects approaching the antibody titer measurements on the same three dates, but not the same time after full vaccination. Since the primary aim was i.a. to compare the antibody titers between measurements, we did not want to introduce another factor into analysis (so as not to complicate the reception of the results even more). Since we noted in Poland a specific months with vaccinations with specific types of vaccines, we looked at the time after the second vaccination with respect to the vaccine type (cf. lines 230-241).
I noticed that the previous reviewer had recommended to describe the study workflow. The revised version includes a table describing the allocation of participants, but not the distribution of the time elapsed from vaccine administration, as well as from natural infection.
The Authors divided the time variable in 3 groups (based on what considerations, since the distribution was not examined?)
Unfortunately, our cohort was varied and from the first study we had people both after the first doses and after being fully vaccinated with one of the 4 available vaccines. The diversity results from the schedule of the availability of individual vaccines in our country. In Poland, the BNT162b2 Comirnaty preparation is available from December 27, 2020 and it is a preparation that performs most vaccinations. Vaccinations with mRNA-1273 Spikevax began in Poland on January 20, 2021, with Vaxzevria (AstraZeneca) from February 12, 2021 and single-dose Johnson & Johnson vaccine was introduced for use on April 14, 2021. for all participants, at the same time point (between the date of blood collection and the date of injection), especially since the tests of UMK employees were group and open to everyone. The time taken for individual participants to attend each stage of the study is also mentioned in the discussion when discussing individual vaccines (e.g., line 472-474 or line 521-524).
Hence, we have made an indirect attempt to normalize the vaccinating time factor by dividing it into subgroups (line 233-238). The division into particular groups was made in accordance with the available literature data and our previous published studies. The time factor related to the dynamics of changes in the level of antibodies was also taken into account when determining the dates of subsequent blood donations. The available data and our previous observations show that around 14-21 days after full vaccinations we have to deal with maximum antibody levels in the case of mRNA vaccines, and around 40-60 days in the case of vector vaccines. After this time, we observe drops in the level of antibodies, which are most intense in the first 3-5 months, depending on the vaccine.
In our study the vaccine types were not evenly distributed across groups. This confirms what we mentioned before: in Poland a specific months with vaccinations with specific types of vaccines Figure 1 shows that in the first group (up to 14 days after the full vaccination) the vaccine types were quite evenly distributed. The situation changes in group 2 (mainly ChAdOx1 nCoV-2019 and JNJ-78436735) and in group 3 (mainly BNT162b2 ) and confirms the association between both factors.
Then, quite surprisingly, they declared that the variable was not taken into account just because it was significantly associated - see lines 198-204 and 254-257:
It is known, that if two explanatory variables are highly correlated (which is indeed show in the paragraph cited by the Reviewer) - in our case time after the full vaccination is strictly associated with the type of the vaccine – this is not a desirable situation to include both of these factors into analysis as predictors for the dependent variable (the antibody titer). In most of statistical procedures it is even incorrect and might adversely affect the result of the analysis (e.g. in linear regression models). The inference concerning one variable can be extended to the other correlated variable after obtaining the results. Moreover, we wanted to compare the types of the vaccine, and if we would consider them in groups defined by the time after vaccination, it would not be correct (since we do not have an even representation of types of vaccines in groups and would not have enough data to compare).
‘Since such an association between the group and the vaccine type was statistically significant, we decided to carry out further analyses without taking the group into account, only considering the type of vaccine, mining the relationship between the time of vaccination (group) and the vaccine type’. Based on their own declaration, thus, the results may be heavily biased, because a significant predictor was not considered in the analyses.
We suspect that the Reviewer's interpretation of the Pearson's chi-squared test of independence might not be correct. The above cited paragraph only shows that we can assume the dependency between the type after full vaccination and the vaccine type (in this meaning the Pearson's chi-squared test is significant – we reject its null hypothesis of independence). But we nowhere state that not the time after full vaccination is a significant predictor (as we stated there is no need to analyse it, since we have another factor correlated with time, more interesting for us).
By the way, in other instances the authors state that the methods employed have been changed during the course of the analyses, depending on findings – see for instance lines164-166, 194-197, 207-210. This looks quite unusual.
This is a standard procedure that before the main analyses, the exploratory data analysis has to be performed. We particularly looked at the numbers of observations in various groups defined by the factors of our interests, and at the assumptions for the parametric procedures (such as analysis of variance). No one should interpret the results of analysis which, during the model diagnostics, occurs to be not well fitted or when the assumptions are not satisfied. This happened (lines 293-298) in case of the analysis which was our first choice to gather primary factors together. However we decided to cite the results, pointing out that they are not necessarily reliable. We then seized to nonparametric methods (and further post-hoc tests), available in appropriate R package, which we employed consequently in further steps of analysis for all factors of our interest.
As a conclusion, we would like thank the Referee for a deep insight into our study. We would also like to emphasise the fact that the choice of the type of procedure is strictly dependent on many factors, i.e. : the human analyst and his/her preferences, the types of explanatory and the response variables, etc. The methods we chose were preceded by many weeks of research of data at our disposal and were not selected randomly but in accordance to our best knowledge and experience. We are aware that it is very difficult to evaluate the results and research that was not carried out by itself, however we hope that our explanations convince the Reviewer that the manuscript is worth publishing in this journal.